# LC-MS/MS Analysis Elucidates a Daily Rhythm in Orexin A Concentration in the Rat Vitreous Body

**DOI:** 10.3390/molecules26165036

**Published:** 2021-08-19

**Authors:** Lukasz Chrobok, Sylwia Bajkacz, Jasmin Daniela Klich, Marian Henryk Lewandowski

**Affiliations:** 1Department of Neurophysiology and Chronobiology, Institute of Zoology and Biomedical Research, Jagiellonian University in Krakow, Gronostajowa 9 Street, 30-387 Krakow, Poland; klich.jasmina@gmail.com (J.D.K.); marian.lewandowski@uj.edu.pl (M.H.L.); 2Department of Inorganic Chemistry, Analytical Chemistry and Electrochemistry, Faculty of Chemistry, Silesian University of Technology, Krzywoustego 6 Street, 44-100 Gliwice, Poland; 3The Biotechnology Centre, Silesian University of Technology, Krzywoustego 8 Street, 44-100 Gliwice, Poland

**Keywords:** orexin, vitreous body, LC-MS/MS, circadian clock

## Abstract

Orexins are two neuropeptides synthesised mainly in the brain lateral hypothalamic area. The orexinergic system provides arousal-dependent cues for a plethora of brain centres, playing a vital role in feeding behaviour, regulation of the sleep–wake cycle and circadian rhythms. Recently, orexins were found to be produced in the retina of an eye; however, their content in the vitreous body and possible daily pattern of expression have not yet been explored. In this manuscript, we describe the development and validation of a liquid chromatography with tandem mass spectrometry (LC-MS/MS) method designed for quantitative bioanalysis of orexin in the rat vitreous body. Orexin was extracted from vitreous body samples with a water:acetonitrile:formic acid (80:20:0.1; *v*/*v*/*v*) mixture followed by vortexing and centrifuging. Separation was performed on a reverse-phase HPLC column under gradient conditions. Orexin was analysed via multiple-reaction monitoring (MRM) in the positive electrospray mode. The total analysis time for each sample was less than 5.0 min. Once the method was fully optimised, it was then validated, following the 2018 FDA guidance on bioanalytical method validations. The calibration curves for orexin (1–500 ng/mL) were constructed using a linear regression with a 1/x^2^ weighting. The lower limit of quantitation for orexin was 1.0 pg/mL for the vitreous body. Intra-day and inter-day estimates of accuracy and precision were within 10% of their nominal values, indicating that the method is reliable for quantitation of orexin in the rat vitreous body. From the physiological perspective, our results are the first to show daily rhythm of orexin synthesis by the retina with possible implications on the circadian regulation of vision.

## 1. Introduction

The rotation of Earth around its axis within a period of 24 h evokes pronounced cyclic environmental alterations of day and night. Living organisms had to adapt to these rhythmic changes, and the ability to anticipate them created a huge evolutionary advantage. Thus, nearly all life on Earth possesses intrinsic timekeeping mechanisms called circadian clocks. In mammals, the master clock is localised in the hypothalamic brain structure—the suprachiasmatic nuclei (SCN) [1,2]. However, accumulating evidence shows the existence of local clocks in other brain areas and peripheral tissues, such as the liver, kidney, lungs [3,4,5,6], or the retina of an eye [7,8,9]. Due to a modern 24/7 lifestyle, shift work or during the course of jet lag, body clocks desynchronise, leading to severe civilisational health problems, including obesity, cardiovascular diseases and some types of cancer [10]. Thus, understanding circadian rhythmicity requires further thorough exploration, including the involvement of new analytical tools for chronobiological investigation.

Orexins are two neuropeptides (orexin A and B, OXA and OXB, respectively) synthesised predominately in the lateral hypothalamic area as a product of proteolysis of their common precursor—prepro-orexin [11,12,13]. The molecular masses of OXA (33 amino acids) and OXB (28 amino acids) are 3562 and 2937 Da, respectively. In contrast to OXB, OXA contains four cysteine residues that form two intrachain disulphide bonds. In the brain, the orexinergic system has been hypothesised to act as the hands of the master clock, providing timed arousal-dependent cues for the plethora of neuronal centres. Due to its reciprocal connections with the SCN, the neuronal activity of orexinergic neurons in rodents follows the light–dark cycle, with higher activity seen during the behaviourally active night [14,15]. It has been additionally shown that orexins are produced and act locally in the retina of rodents and humans [16,17,18,19,20]. A recent report demonstrated retinal orexins to control the pupillary light reflex [21]. However, the temporal coordination of orexin release in the retina on the daily timescale remained elusive.

To verify the above hypotheses, it is necessary to develop an analytical method for the determination of orexin in the vitreous body, a gel-like fluid filling the vitreous chamber of the eyeball ranging from the lens to the retina. Due to the close proximity to retinal cells, the vitreous body not only provides a clear pathway for light to reach the photoreceptors but also creates a reservoir for many substances synthesised by the retina [22,23]. Only three analytical methods for the determination of orexin have been published to date. Two of these methods for quantification of OXA in the cerebrospinal fluid are based on liquid chromatography with tandem mass spectrometry (LC-MS/MS) analysis and standard radioimmunoassay [24,25]. In the third paper, the authors focus on research on reducing adsorption to improve recovery and in vivo detection of OXA and OXB by microdialysis with LC-MS [26]. These methods were applied only for determination of orexin in the cerebrospinal fluid, and no simultaneous analysis of the vitreous body has been described hitherto.

Therefore, the aim of the study was to detect and quantify OXA content in the rat vitreous body with the use of LC-MS/MS analytical methods. Here, we show that OXA may be reliably detected from vitreous body samples, which proves their local expression in the retina. Additionally, our study provides compelling evidence for daily fluctuations in the OXA content in the eye, which opens new pathways for further physiological and pharmacological studies.

## 2. Materials and Methods

### 2.1. Chemicals and Reagents

Orexin A (OXA) (purity 97%) was purchased from Bachem (Bubendorf, Switzerland). Acetonitrile and methanol (HPLC-MS grade), water and formic acid (HPLC grade) were purchased from Merck (Darmstadt, Germany). The stock solution of OXA was prepared in methanol at a 1.0 mg/mL concentration. Stock solutions were further diluted with methanol to obtain OXA working solutions at several concentration levels.

Calibration standards (CS) and quality control (QC) samples in the rat vitreous body were prepared daily by diluting the corresponding working solutions with an OXA-free (blank) rat vitreous body. Before preparing the CS and QC samples, the rat vitreous body applied as a blank sample was examined to ensure that it was free of OXA. The CS and QC samples were prepared on each analysis day with the procedures described below. The concentrations of the CS were 1, 10, 20, 50, 100, 200, 350 and 500 pg/mL of OXA in the rat vitreous body. The QC samples in the rat vitreous body were prepared at three concentrations: lower limit of quantitation (LLOQ): 1 pg/mL; low concentration quality control (LQC): 10 pg/mL; middle concentration quality control (MQC): 100 pg/mL; and high concentration quality control (HQC): 500 pg/mL. All stock, standard working and QC working solutions were stored at 4 °C and were brought to room temperature before use. Quantitative analysis of the OXA was performed for the rat vitreous body samples; therefore, the validation of the method was also performed for this matrix (CS and QC samples prepared in the rat vitreous body).

### 2.2. Ethical Approval

Animals were maintained and used according to Polish regulations and the European Communities Council Directive (86/609/EEC). All procedures were approved by the Local Ethics Committee in Krakow and all efforts were made to minimise the number of animals used and their suffering.

### 2.3. Animals

This study was performed on 20 adult (8–10 week old) Sprague–Dawley rats of both sexes. Rats were bred and housed in the Animal Facility at the Institute of Zoology and Biomedical Research, Jagiellonian University in Krakow, three to six per cage at 23 ± 2 °C and 67 ± 3% relative humidity. Animals were provided with ad libitum access to food and water and kept under standard 12:12 h light–dark (LD) cycle. All procedures in darkness were performed under dim red light, not visible to rodents.

### 2.4. Tissue Collection

Rats were deeply anaesthetised with isoflurane inhalation (2 mL per chamber) and intracardially injected with an overdose of sodium pentobarbital (100 mg/kg body weight; Biowet, Pulawy, Poland). Following this, the eyes were enucleated, rinsed in distilled water and quickly dried with a disposable tissue paper. Then, the eyes were hemisected with a fresh surgical blade, and the vitreous body (20 µL) was collected in a fresh test-tube. Samples from both eyes were pooled together, flash frozen over dry ice and kept at −80 °C.

Culls were performed at four daily time points (n = 5 rats per group), at Zeitgeber time (ZT) 1, 7, 13 and 19, where ZT0 means the time of lights-on and ZT12 means lights-off. Procedures at ZT1 and ZT7 were performed in ambient light (~300 lux), whereas those at ZT13 and ZT19 were carried out under dim red light.

### 2.5. Sample Preparation

A 10 μL sample aliquot (rat vitreous body) was placed in a 1.5 mL microcentrifuge tube. The sample was extracted with 90 µL of mixture water:acetonitrile:formic acid (80:20:0.1; *v*/*v*/*v*) by vortex mixing for 10 min. The supernatant was collected after centrifugation at 15,000× *g* for 10 min. Ten microlitres of the supernatant was injected into the LC-MS/MS system for analysis.

### 2.6. LC-MS/MS Analysis

LC-MS/MS analysis was performed on a Dionex UHPLC system (Dionex Corporation, Sunnyvale, CA, USA) coupled to a QTRAP 4000 triple quadrupole linear ion trap mass spectrometer equipped with an electrospray ionisation (ESI) interface (Applied Biosystems/MDS SCIEX, Foster City, CA, USA). The LC system includes an autosampler, a binary pump, a compartment with a thermostable column area and a variable wavelength detector. The data were acquired using Analyst software, Version 1.4.

OXA analysis was carried out on a Kinetex C18 analytical column (75 × 2.1 mm; 2.6 µm) with column temperature set at 30 °C. The mobile phase consisted of 0.1% formic acid in water (mobile phase A) and acetonitrile (mobile phase B). Sample separation was conducted under gradient conditions with a flow rate of 0.8 mL/min. The initial elution composition was 90% (*v*/*v*) mobile phase A, followed by a linear gradient up to 65% (*v*/*v*) mobile phase A at 3 min and returning back to initial composition at 3.1 min. The column equilibration time was 1.9 min, leading to a total run time of 5 min. The injection volume was 5 μL. The autosampler’s temperature was set at 10 °C.

To select the MS/MS parameters, a standard solution of OXA at concentration 500 ng/mL was infused into the mass spectrometer using a Harvard syringe pump at a flow rate of 10 µL/min. Finally, multiple reaction monitoring (MRM) mode was applied for the detection of OXA by monitoring the transitions: MRM1 (quantifier) *m*/*z* 713.3 → 858.6 and MRM2 (qualifier) *m*/*z* 713.3 → 854.1 with a dwell time of 250 ms for each transition. The collision energy was optimised to 22 V and 25 V for MRM1 and MRM2, respectively. Other compound-dependent parameters (declustering potential (DP), collision energy (CE), entrance potential (EP) and collision cell exit potential (CXP)) were as follows: 100 V; 10 V; 7 V. The IonSpray voltage was set at 5500 V in the positive ionisation mode. The source temperature was 500 °C. Nitrogen (nebulising gas) pressure, turbo spray gas pressure, curtain gas and collision gas pressure were set at 50 psi, 40 psi, 20 psi and 5 psi, respectively.

### 2.7. Method Validation

Selectivity: Selectivity assessment was performed in chromatograms of blank rat vitreous body samples compared to those with an OXA spike to explore the potential interference with the analyte.

Linearity and LLOQ: The linearity test was conducted by injecting eight calibration standard solutions of concentrations ranging from 1 to 500 pg/mL of OXA in the rat vitreous body and fitted to *y* = bx + c by a weighting factor of 1/*x*^2^. The lower limit of quantification (LLOQ) was defined as the lowest calibration curve concentration, at which the deviation of accuracy (RE) was within ±20% and precision (RSD) < 20%. The signal-to-noise ratio (S/N) was more than 10.

Precision and accuracy: Precision and accuracy were calculated as relative standard deviation (%RSD) and relative error (%RE) from the theoretical/nominal values, respectively. Three replicates of each QC sample, namely, LLOQ = 1 pg/mL, LQC = 10 pg/mL, MQC = 100 pg/mL and HQC = 400 pg/mL, were analysed on the same day (inter-day) and on different days (intra-day) with the calibration curve to determine the precision and accuracy of the method. The deviation in the precision (%RSD) and accuracy (%RE) values was limited to ≤15% for the QC samples except for LLOQ samples, where it was limited to ±20% (%RSD and %RE).

Matrix effect and recovery: To evaluate the matrix effect (ME), blank matrices were extracted and spiked with the QC samples (post-spike sample). The ME was calculated by comparing the peak area of OXA in post-spike samples with that in external (unextracted) samples at the same concentration. The extraction recovery (ER) of OXA from the rat vitreous body was evaluated by comparing the mean area of the extracted QCs with that of samples at the same concentration obtained by spiking the standard solution with the extracted blank matrix samples.

Stability: The stability of the analytes at different storage conditions was also assessed at four concentration levels (LLOQ, LQC, MQC and HQC). Short-term and long-term stabilities were evaluated for spiked rat vitreous body samples kept in the autosampler for up to 24 h (10 °C), at −80 °C for one week and at −20 °C for one month, respectively. Freeze–thaw stability was also assessed in three freeze–thaw cycles. Spiked QC samples were kept frozen at −80 °C overnight and thawed at room temperature the following day. The third day QCs were thawed, prepared and analysed, as described previously. All analysed samples were compared to freshly prepared matched QC samples.

Carry-over effect: The carry-over effect was evaluated by injecting blank rat vitreous body samples and methanol, respectively, after the analysis of samples spiked at 1000 pg/mL OXA.

Dilution integrity: Dilution integrity was demonstrated using samples spiked at a concentration of 1000 pg/mL, which were diluted to 1/10 of their concentration with a blank rat vitreous body. Dilution integrity was determined with six replicates.

### 2.8. Statistics

First, data triplicates were averaged for each animal (n = an average of three analyses from one vitreous body sample). SD was measured for these animal-averaged values, thus demonstrating inter-subject differences. Statistical testing based on nM concentration values was performed in Prism 7 (GraphPad Software, San Diego, CA, USA) with ordinary one-way ANOVA. Normality of the distribution was assessed with the Kolmogorov–Smirnov test with Dallal–Wilkinson–Lilliefor p-values. Data were then transformed to relative orexin content (%ZT1) by calculating the % value for each data point in relation to ZT1 average (set at 100%). Next, relative values were tested for rhythmicity in CircWave v1.4 software (developed by Dr Roelof Hut, University of Groningen, The Netherlands, http://www.euclock.org/results/item/circ-wave.html (accessed on 23 June 2021)). All data are presented as mean ± standard deviation (SD), and *p* < 0.05 was deemed significant.

## 3. Results and Discussion

### 3.1. LC-MS/MS Method Development

The MS/MS parameters were optimised to produce the maximum response for the orexin standard. The MS signal was optimised in full scan mode in the *m*/*z* range of 100–900 under positive ionisation. The protonated molecular ion [M + 5H]^5^^+^ with *m*/*z* 713 was identified as the major peak for the selected peptide. The two most intense fragment ions observed were the ions with *m*/*z* 858.6 and *m*/*z* 854.1. The ion with *m*/*z* 858.6 was selected since it was produced in relatively greater abundance, which was also reported in previous studies [25]. The analyte-dependent parameters, especially collision energy, were optimised for each transition in order to obtain the most sensitive method (Section 2.6).

The HPLC conditions were optimised to eliminate interferences from the matrix and to elute the analyte within a short run time (5.0 min). To obtain the best chromatographic conditions, the mobile phase composition, type of column, column temperature and flow rate were judiciously selected. The Kinetex C18 column was found to be optimal compared with the tested columns, including Chromolith^®^ Fast Gradient Monolithic C18e, Hypersil Gold and Kinetex F5 due to the unique, superficially porous particles and 2.6 μm particle size providing high resolution of the tested analyte from matrix components. In addition, this column enabled a robust symmetrical peak to be obtained and a more sensitive method to be used compared to other columns.

The feasibility of using various mixtures of solvents, such as acetonitrile and methanol, and different additives, such as ammonium acetate, ammonium formate, acetic acid and formic acid, with a variable pH range of 2.5–5.5, along with altered flow rates (in the range of 0.3–1.0 mL/min), was tested for the complete chromatographic resolution of OXA and matrix components. The optimal mobile phase was found to consist of acetonitrile and water containing 0.1% (*v*/*v*) formic acid (pH = 3.0). A gradient elution was applied to improve the separation of OXA from the eluting interferences and to facilitate efficient ionisation of the analyte under electrospray conditions. Acetonitrile rather than methanol was chosen as the organic modifier because acetonitrile led to lower background noise and resulted in the best resolution. During the early stages of method development, a small amount of acidic modifier (0.1% (*v*/*v*) formic acid) was added to the mobile phase to improve the MS/MS response of the analyte of interest. Moreover, the effects of column temperature were investigated in the range of 20–45 °C. The retention time decreased slightly, and no change in peak shape was observed with increasing temperature. Therefore, the temperature was maintained at 30 °C. The developed LC conditions employed in the present method resulted in a retention time of 3 min for OXA (Figure 1). Additionally, representative chromatograms obtained in MRM acquisition mode for OXA at three concentration levels are shown in Appendix A.

Mixtures of methanol, acetonitrile, water, formic acid and acetic acid were used for the extraction under acidic to neutral pH conditions. The lowest recovery of OXA, 10%–30%, was obtained for pure water. Adding acetonitrile to samples increased the signal for the studied peptide. We investigated the effect of acetonitrile added at different concentrations (5%−100%). Optimal acetonitrile concentrations were from 15% to 25%. Increasing the amount of organic phase in the extraction mixture was found to significantly lower its extraction efficiency. Extraction with pure acetonitrile gave recoveries ranging from 50% to 60%. The addition of acetonitrile to water in a 1:1 volume ratio increased the recovery of OXA by 20% compared to pure acetonitrile. The highest recovery was obtained for the mixture of water:acetonitrile:formic acid (80:20:0.1; *v*/*v*/*v*). The conversion of acetonitrile to methanol caused the extraction yield to drop by 30% due to the higher polarity of the mixture. The use of the acetic acid instead of formic acid resulted in an increase in OXA recovery (70%). Summarising, a solvent mixture of water:acetonitrile:formic acid (80:20:0.1; *v*/*v*/*v*) was identified to affect maximal extraction of orexin from vitreous body samples.

### 3.2. Method Validation

This method was validated and included all the performance criteria needed, such as selectivity, linearity, sensitivity, accuracy, precision, recovery, matrix effect, carry-over effect and dilution integrity.

Selectivity: For the blank vitreous body, there was a minor peak observed with a peak area of less than 20% of the LLOQ area. Based on the consistently low intensity of this signal, it was determined that the endogenous peak would not substantially affect the accurate quantitation of OXA. Exemplarily, MRM chromatograms of the extract of the vitreous body spiked with 1 pg/mL OXA and in the extract of the blank vitreous body are presented in Figure 2.

Linearity and LLOQ: The calibration curves showed good linearity (r > 0.9921) over the tested concentration ranges of OXA in the rat vitreous body samples. The method showed good sensitivity at the level of 1 pg/mL (S/N > 10) for OXA in the rat vitreous body samples, which were sufficient to study the orexin content in the eye.

Precision and accuracy: The calculated results of intra-day and inter-day precision as well as accuracy data for OXA (at four concentration levels of QC samples) in the rat vitreous body are represented in Table 1. The intra-day precision (%RSD) was found in the range of 2.8% to 5.2%, and accuracy (%RE) was in the range of −6.8% to −2.0%. Similarly, the inter-day precision was found in the range of 4.4% to 6.5%, and accuracy was in the range of −9.1% to −5.7%, which indicates that the developed LC-MS/MS method is suitable for the analysis of OXA in rat vitreous body samples.

Matrix effect and recovery: Reasonable values of ME ranging from 102.1% to 110.2% were obtained for quality control samples that were spiked at four concentrations: LLOQ, LQC, MQC and HQC. Adequate ER values ranging from 95.4% to 98.3% were obtained for OXA at four concentration levels (Table 1).

Stability: The stability data are presented in Table 2. In all the stability tests, the analytes showed good stability with RE values ranging from −9.8% to −2.9%, and CV values were <8.3% at different storage conditions (autosampler, short-term and long-term stability, freeze–thaw stability) suggesting no significant degradation.

Carry-over effect: Carry-over in the first double blank after ULOQ determined in three runs for OXA was 1.3% to 6.6% of the LLOQ mean peak area. The acceptance criteria were met. Carry-over is expected not to have any impact on study sample results. Nevertheless, carry-over was monitored and evaluated during study sample analysis in each batch by injecting double blanks/blanks after ULOQ.

Dilution integrity: One tenfold dilution of a 1000 ng/mL dilution QC sample (n = 6) resulted in accuracy and precision of 1.6% and 2.5%, respectively, for the rat vitreous body samples, thus meeting the acceptance criteria.

All results obtained during method validation met the acceptance criteria of the US-FDA Guideline for Bioanalytical Validation (2018) [27].

### 3.3. Method Applicability

The method described above was used to quantify OXA concentrations in the biological samples across the daily cycle. Twenty samples of the vitreous body were collected at four time points across 24 h (ZT1, 7, 13, 19, where ZT0 means the time of lights-on and ZT12means lights off). Each sample was analysed in triplicate, and the obtained results were averaged for each animal. Figure 3 shows the MRM chromatogram of the extract of the rat vitreous body after LC-MS/MS analysis. The average OXA concentration in the vitreous body for all samples amounted to 13.09 ± 3.5 nM (n = 20), which stands in the physiological range of OXA action upon orexin receptors (IC_50_ = 20 and 38 nM for orexin receptors 1 and 2, respectively [11]).

However, further analysis proved that OXA concentration is subject to daily variation (*p* = 0.0002, n = 20, one-way ANOVA; Figure 4A). It stayed in relatively high values during the night to the beginning of the light phase, with a notable drop in the middle of the day (reaching 53.02 ± 9.1% of ZT1 mean). Next, the rhythmicity of this daily pattern was assessed by sine curve fitting. Indeed, the analysis proved the distribution to be rhythmic (*p* = 0.049, n = 20, CircWave; Figure 4B), with an acrophase of the rhythm at ZT19.9. This dataset shows that OXA may be reliably quantified in the rat vitreous body samples, showing physiologically relevant concentrations. Additionally, it shows a significant daily rhythmicity of OXA content peaking in the middle of the night and exhibiting a notable drop during the behaviourally quiescent day. This daily pattern resembles the brain orexin rhythm in the lateral hypothalamus and subcortical visual structures [15,28,29].

## 4. Conclusions

The present study reports the development and validation of a sensitive and specific LC-MS/MS method for the quantification of OXA in the rat vitreous body. Advantages include a chromatography run time of 5 min and the use of a small volume of sample (10 μL) with improved analytical sensitivity (1 pg/mL). The method was designed according to the current FDA bioanalytical guidance requirements as fit for purpose. Moreover, this report is the first to provide a quantitative description of OXA presence in the eye vitreous body in concentrations sufficient to functionally activate orexin receptors. Importantly, retinal-derived orexin content followed a daily pattern, with a peak in the middle of the behaviourally active night. This suggests either the retinal clock-dependent expression of the prepro-orexin gene or regulation of expression/secretion via the light–dark cycle. These findings, possible due to the utilisation of our LC-MS/MS method, provide valuable physiological background to the orexinergic modulation of early visual processing.

## Figures and Tables

**Figure 1 molecules-26-05036-f001:**
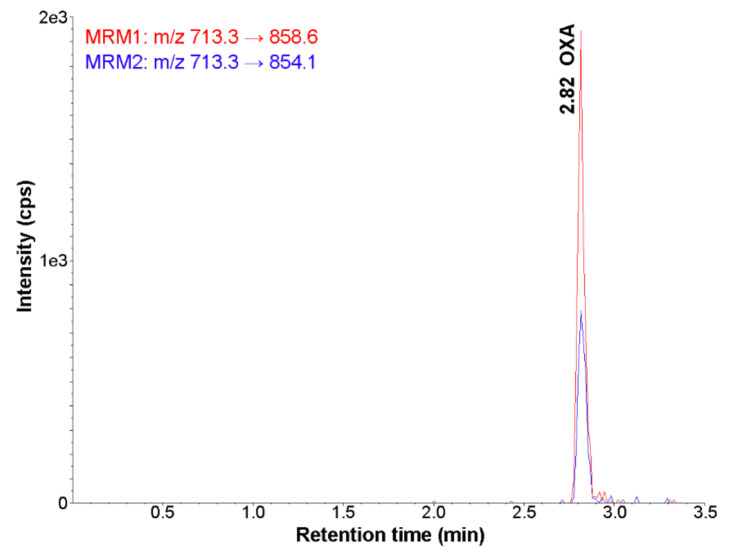
Representative MRM chromatogram of orexin (standard solution at concentration 100 pg/mL) (UHPLC-MS/MS method).

**Figure 2 molecules-26-05036-f002:**
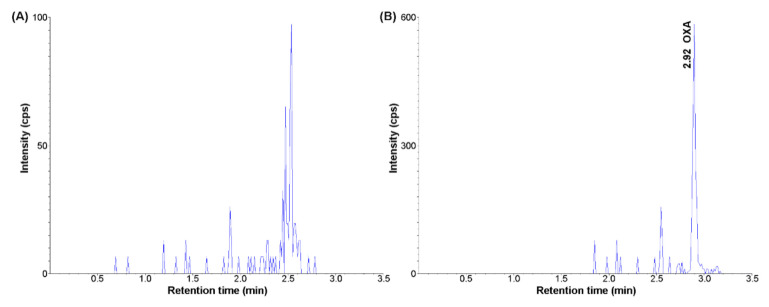
Representative MRM chromatograms of orexin A: (**A**) blank vitreous body; (**B**) blank vitreous body spiked with orexin A (1 pg/mL) (UHPLC-MS/MS method).

**Figure 3 molecules-26-05036-f003:**
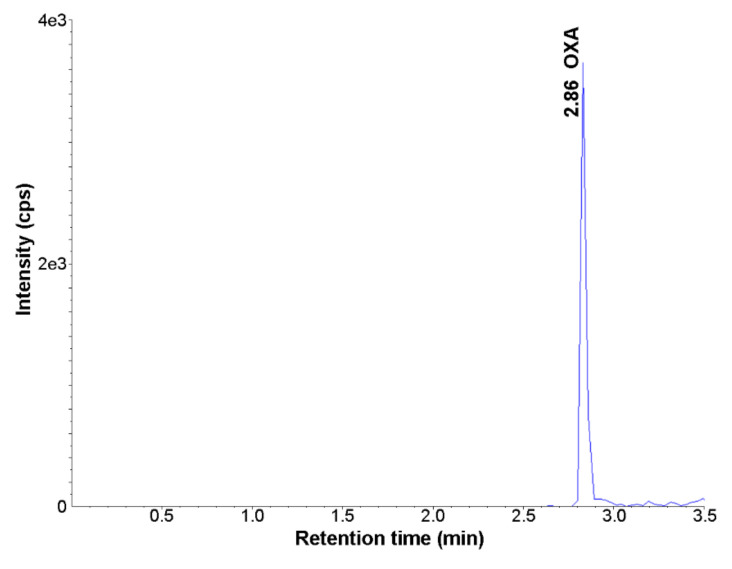
Representative MRM chromatogram of orexin A in the rat vitreous body sample (MRM1; *m*/*z* 713.3 → 858.6) (UHPLC-MS/MS method).

**Figure 4 molecules-26-05036-f004:**
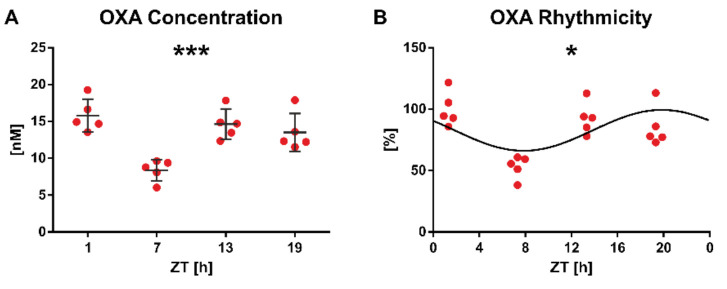
Orexin A (OXA) exhibits daily rhythmicity in the rat vitreous body samples. (**A**) Daily variation in concentration assessed by one-way ANOVA (*** *p* = 0.0002, n = 20). (**B**) Daily rhythmicity studied with sine wave fitting (* *p* = 0.049, n = 20; CircWave). Each data point demonstrates an average of three measurements obtained from one vitreous body collected from one animal. %—relative OXA concentration to ZT1 average. ZT—Zeitgeber time. Data are presented as a mean ± SD.

**Table 1 molecules-26-05036-t001:** Intra-day and inter-day precision and accuracy, extraction recovery and matrix effect for OXA-spiked rat vitreous body (n = 6).

Concentration Added (pg/mL)	Precision RSD (%)		Accuracy RE (%)		Matrix Effect ME (%)	Extraction Recovery ER (%)
	Intra-Day	Inter-Day	Intra-Day	Inter-Day		
1	5.2	6.5	−6.8	−9.1	110.2	95.4
10	3.7	5.2	−3.6	−8.3	107.5	97.1
100	3.1	5.0	−2.1	−7.3	107.2	97.8
400	2.8	4.4	−2.0	−5.7	102.1	98.3

**Table 2 molecules-26-05036-t002:** Stability results of OXA in rat vitreous body in different conditions (n = 6).

Concentration Added (pg/mL)	Autosampler Stability		Short-Term Stability		Long-Term Stability		Freeze–Thaw Stability	
	RSD (%)	RE (%)	RSD (%)	RE (%)	RSD (%)	RE (%)	RSD (%)	RE (%)
1	4.8	−6.4	5.8	−6.9	5.2	−9.8	8.3	−9.1
10	4.4	−5.6	4.8	−6.3	4.4	−8.4	3.2	−8.9
100	2.8	−5.5	4.0	−4.5	3.1	−5.7	2.5	−5.7
400	1.6	−3.2	2.3	−3.9	2.1	−2.9	1.9	−4.5

## Data Availability

The data that support the findings of this study are available from the corresponding authors upon reasonable request.

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
