# Peer review of "LC-MS/MS Analysis Elucidates a Daily Rhythm in Orexin A Concentration in the Rat Vitreous Body"

_molecules, 2021, doi:10.3390/molecules26165036_

Round 1

Reviewer 1 Report

  • Introduction: Add some physicochemical properties of OXA.
  • Chemicals and reagents: add the purity of OXA standard.
  • Results and discussion: Give some results about method development (mobile phase components, flow rate, column selection). Did you check with isocratic mobile phase?
  • Results and discussion: Explain extraction according to the physicochemical properties of OXA. Explain the pH used in the extraction solvent. Did you check with another extraction solvent?

Author Response

Dear Editors and Reviewers of Molecules

Thank you for your review of our manuscript.

We have corrected the manuscript according to your comments. All the changes made are indicated using the red font and the yellow highlighter in the corrected version of the manuscript. In reply to your remarks, we would cite the remarks of the reviewers and respond to them one by one. We would like to express our gratitude for the comments on this manuscript, which we believe greatly improved its clarity and overall quality.

Sincerely,

Łukasz Chrobok

Sylwia Bajkacz

Reviewer 2 Report

This manuscript has provided certain data for their study on orexin A concentration, but it is lack of explanation on data analysis process, and need more discussions on the physiological details of orexin A to support their conclusion.  Major revision is required to address these issues.  More details of these issues are listed below.

  1. Why didn’t show any chromatogram from UHPLC?
  2. Figure 1 shows very little information, should add the chromatograms of orexin at three different concentrations.
  3. For each vitreous body, how many measurements did the author use? How did the authors calculate SD? In Figure 4, at each time points, what do the five red dots mean? Are all measurements mixed into one sample pool or separated according to different vitreous body?
  4. In Figure 4, it seems like the concentration around 19th hour have a very different deviation point from the other three data point, so how to explain that outliner?
  5. Again in Figure 4B, the unit of y axis is % for what? How did the author calculate this value?
  6. It is really difficult to convince me that the data points in Figure 4 can be fitted to sine wave and represent daily rhythmicity. What is the biological mechanism behind the orexin in vitreous body? How did the authors choose that four time points? Did the 1, 7, 13, and 19th hour really related to physiological concertation of orexin?

Author Response

(The authors gave the same response as above.)

Round 2

Reviewer 2 Report

The revision is much better now.  I do hope this manuscript can provide more discussion on physiological background and mechanism information related to in vivo study, but the present form is good enough to be published.